# A Cytotoxic Porphyrin from North Pacific Brittle Star *Ophiura sarsii*

**DOI:** 10.3390/md19010011

**Published:** 2020-12-29

**Authors:** Antonina Klimenko, Robin Huber, Laurence Marcourt, Estelle Chardonnens, Alexey Koval, Yuri S. Khotimchenko, Emerson Ferreira Queiroz, Jean-Luc Wolfender, Vladimir L. Katanaev

**Affiliations:** 1Translational Research Center in Oncohaematology, Department of Cell Physiology and Metabolism, Faculty of Medicine, University of Geneva, CH-1211 Geneva, Switzerland; antonina.klimenkot@unige.ch (A.K.); Estelle.Chardonnens@etu.unige.ch (E.C.); alexey.koval@unige.ch (A.K.); 2School of Biomedicine, Far Eastern Federal University, 690090 Vladivostok, Russia; khotimchenko.ys@dvfu.ru; 3School of Pharmaceutical Sciences, University of Geneva, CMU—Rue Michel-Servet 1, CH-1211 Geneva, Switzerland; Robin.Huber@unige.ch (R.H.); laurence.marcourt@unige.ch (L.M.); 4Institute of Pharmaceutical Sciences of Western Switzerland, University of Geneva, CMU—Rue Michel-Servet 1, CH-1211 Geneva, Switzerland

**Keywords:** breast cancer, *Ophiura sarsii*, targeted therapy, cytotoxicity, porphyrin, photodynamic therapy

## Abstract

Triple-negative breast cancer (TNBC) represents the deadliest form of gynecological tumors currently lacking targeted therapies. The ethanol extract of the North Pacific brittle star *Ophiura sarsii* presented promising anti-TNBC activities. After elimination of the inert material, the active extract was submitted to a bioguided isolation approach using high-resolution semipreparative HPLC-UV, resulting in one-step isolation of an unusual porphyrin derivative possessing strong cytotoxic activity. HRMS and 2D NMR resulted in the structure elucidation of the compound as (3*S*,4*S*)-14-Ethyl-9-(hydroxymethyl)-4,8,13,18-tetramethyl-20-oxo-3-phorbinepropanoic acid. Never identified before in Ophiuroidea, porphyrins have found broad applications as photosensitizers in the anticancer photodynamic therapy. The simple isolation of a cytotoxic porphyrin from an abundant brittle star species we describe here may pave the way for novel natural-based developments of targeted anti-cancer therapies.

## 1. Introduction

Despite the enormous effort, cancer remains one of the leading causes of disease-related mortality. To overcome this persistence, targeted anticancer therapies are in need and are being actively developed. The targeting of the tumor while maximally sparing the healthy tissues can be achieved chemically, i.e., by developing drugs selective for oncogenic molecules or pathways, or physically. As an example of the former, drugs attacking the oncogenic Wnt signaling pathway are being actively developed by pharma companies and academia [1,2]. Aberrant Wnt signaling underlies oncogenic transformation and progression in several cancer types, such as colon cancer or of the deadliest cancer in women, triple-negative breast cancer (TNBC) [3,4]. As an example of the latter, photodynamic therapy (PDT) applies loading (topical or systemic) of tissues with a chemical photosensitizer followed by local laser-based long-wavelength tumor illumination, which energetically activates the photosensitizer to emit active oxygen radicals and destroy the tumor cells [5,6].

In our quest to identify novel Wnt-targeting anti-TNBC compounds, we among other approaches screened extracts from deep-sea invertebrates of the Russian Pacific in a set of assays monitoring Wnt signaling in the cancer cells, identifying *Ophiura irrorata* and other brittle stars collected at the depths of 2.2–3.3 km as a promising source of a powerful anticancer activity [7,8]. Since species collection through deep-sea expeditions necessarily provides a limited material, we decided to investigate whether more easily acquirable Pacific brittle stars could similarly possess interesting anti-TNBC activities. In doing so, we turned our attention to *Ophiura sarsii*.

The brittle star *Ophiura sarsii* Lütken, 1855 [9] (family Ophiuridae, order Ophiurida, class Ophiuroidea, phylum Echinodermata) is wide-spread from Northern Atlantic, over the Arctic, and all the way to Northern Pacific, inhabiting waters from shallow to deep [10]. Extracts from the specimen abundant around the Russky island (Peter the Great Gulf, Sea of Japan) were characterized for the anticancer activities against TNBC cells. While possessing a strong anti-Wnt activity (to be described in detail in a separate article), the extracts also revealed a separate potent cytotoxic activity, which surprisingly was found to be mediated by a porphyrin compound. Not previously identified in the class Ophiuroidea [11,12,13], synthetic porphyrins have found wide applications in the photodynamic therapy (PDT). Our findings may have identified a natural photosensitizer for future PDT applications.

## 2. Results and Discussion

Brittle stars *Ophiura sarsii* were collected near the Russky Island in the Peter the Great Gulf, Sea of Japan, in May 2019. After homogenization, an increasing solvent polarity extraction was performed with heptane, chloroform, ethanol, and water (see Methods). Metabolite profiling of the different extracts by UHPLC-PDA-ELSD-MS (ultra high performance liquid chromatography- photodiode array- evaporative light scattering- mass spectrometry) revealed the presence of abundant apolar compounds in the chloroform and ethanol extracts while polar compounds were found to be scarce in the aqueous extract (Appendix A).

The ethanol extraction produced the highest amount of material (509 mg dry mass), but presented an important number of signals at the beginning of the HPLC chromatogram only detected by the ELSD suggesting the presence of polar compounds or inert marine salts (Appendix A). A procedure to eliminate them was thus performed, whereas the extract was submitted to C_18_ solid-phase extraction (SPE) by elution with water and methanol.

The SPE generated a water fraction containing the polar compounds, most probably inorganic salts (Figure 1A), and a methanol fraction enriched in secondary metabolites of interest (Figure 1B). Both samples were submitted to a cellular screening using the triple-negative breast cancer (TNBC) cell line BT-20 in the TopFlash assay monitoring the level of Wnt signaling, and in the CMV-Renilla assay monitoring the acute cytotoxicity [14]. The assay revealed that the anti-Wnt and the cytotoxic activities were mainly found in the salt-free methanolic SPE fraction of *O. sarsii* (Figure 1C,D).

In order to enrich active metabolites at a larger scale, a liquid–liquid extraction was performed that yielded the butanol fraction (Figure 1E). After confirmation of the anti-Wnt and the cytotoxic biological activities of the butanol fraction (data not shown), the separation conditions of the fraction were first optimized with HPLC-PDA-ELSD using a C_18_ stationary phase. A geometrical transfer of the HPLC analytical conditions to the semipreparative scale using the same stationary phase was performed, and the sample was injected by dry load to improve the chromatographic resolution at a semipreparative scale (see Methods, Figure 2A). The collected fractions were tested for their activity in the TopFlash and CMV-Renilla assays. While the identification of the compound specific anti-Wnt activity will be described in a separate article, we here were attracted by the localization of the acute cytotoxic activity in fraction #14, which possessed a strong dark-blue to black coloration (Figure 2B,C). NMR and HRMS analysis of fraction #14 revealed the presence of a pure compound (**1**) described below.

UV spectra of compound **1** reveal a typical UV absorbance of porphyrins at the wavelengths of UV λ_max_ 204, 293 sh, 409, and 663 nm [15,16,17,18] (Figure 3A). The HRMS spectrum of compound **1** showed a protonated ion at *m/z* 539.2644 [M + H]^+^, (calculated for C_32_H_35_N_4_O_4_, 539.2653, Δ = 1.6 ppm). The ^1^H and edited-HSQC NMR spectra presented typical signal of porphyrin: three deshielded protons and shielded carbon corresponding to aromatics (at δ_H_/δ_C_ 8.83/93.2, H/C-20, 9.57/96.7, H/C-5, 9.75/103.9, and H/C-10) and to C-methyl groups (at δ_H_/δ_C_ 3.25/10.5, H/C-7a, 3.39/10.3, H/C-2a, 3.63/11.2, and H/C-12a). An ethyl group at δ_H_ 1.64 (3H, t, *J* = 7.6 Hz, H3-8b) and 3.73 (2H, q, *J* = 7.6 Hz, H2-8a), a methyl doublet at δ_H_ 1.77 (3H, d, *J* = 7.4 Hz, H3-18a), and a methylene group at δ_H_ 5.12 (1H, d, *J* = 19.4 Hz, H-13b”) and 5.23 (1H, d, *J* = 19.4 Hz, H-13b’) were also observed. These NMR data (see Methods and Appendix A) were very similar to those of pyropheophorbide a [19] except that the signals corresponding to the vinyl group at C-3 were not observed. On the other hand, the signal of an oxygenated methylene group was detected at δ_H_/δ_C_ 5.74/53.9 (H/C-3a). Due to the small amount available, an HMBC experiment could not be recorded, but the ROESY correlations between the different groups (Figure 3B) allowed one to identify compound **1** and determine its relative configuration. The positive optical rotation power of compound **1** ([α]^20^_D_ +350) indicated that absolute configuration was the same than pyropheophorbide a and allowed compound **1** to be fully identified as (3*S*,4*S*)-14-ethyl-9-(hydroxymethyl)-4,8,13,18-tetramethyl-20-oxo-3-phorbinepropanoic acid. Upon chemical synthesis, NMR analysis of this compound has been reported but restricted to ^1^H chemical shifts recorded in a solvent different to ours [20], complicating the direct comparison of NMR data comparison is difficult (Appendix A). The overall yield of the compound was high: 4.2 mg from the 212 mg of the butanol phase (obtained from the 509 mg of the ethanol extract’s dry weight, obtained from 120 g of the wet weight of the collected brittle stars).

The cytotoxic activity of compound **1** was next confirmed in an independent assay. Here, the proliferation of a panel of TNBC cells (BT-20, HCC-1395, HCC-1806, MDA-MB-231, and MDA-MB-468), along with that of a non-TNBC breast cancer line MCF-7 and a non-cancerous breast epithelial line MCF-10A, was monitored in the standard MTT assay (see Methods). The results confirm the broad cytotoxicity of the brittle star porphyrin (**1**) with an IC_50_ around 30 μM (Figure 3C). This was comparable to the cytotoxicity of other porphyrin derivatives described in the literature [15,16,17,21]. Further, this MTT-measured cytotoxicity is similar to that observed in the CMV-Renilla assay in BT-20 cells, which gave the IC_50_ of 37 μM (Figure 2C). Noteworthy, the cytotoxicity of compound **1** against cancerous and non-cancerous breast epithelial cells is similar (Figure 3C). This is typical for photosensitizers applied in the photodynamic therapy (PDT). In PDT, the targeted nature of the treatment is achieved not through selective toxicity of compounds against cancerous cells. Instead, in PDT, tissues are preloaded with the photosensitizer (ideally absorbing in the red-shifted part of the spectrum that penetrates deeper into biological tissues) either systemically or topically, which is followed by targeted laser-based illumination with the correct wavelength. As a result, the photosensitizer is activated, and emits oxygen radicals, toxic for the cells [5,6].

Porphyrins have been identified in several marine living organisms including Echinodermata [13]. However, they were not seen, and were considered absent, in the class of Ophiuroidea [11,12]. Subsequent studies will address whether (3*S*,4*S*)-14-ethyl-9-(hydroxymethyl)-4,8,13,18-tetramethyl-20-oxo-3-phorbinepropanoic acid is synthesized by *O. sarsii* itself, or whether the porphyrin in these brittle stars is an outcome of their dietary preferences. In the latter case, it will be worth investigating whether such dietary acquisition (with potential subsequent in-host modification) of the porphyrin is a local phenomenon, or whether it is as widespread as the northern hemisphere areal of *O. sarsii*. It is worth-mentioning that a similar porphyrin, pyropheophorbide a methyl ester, has been described in the red tide dinoflagellates *Heterocapsa circularisquama* [18]. In the former case, subsequent studies will be directed to address the question of the physiological role the porphyrin plays in *O. sarsii*, which might be linked e.g., with a protective function of the compound.

Interestingly, (3*S*,4*S*)-14-Ethyl-9-(hydroxymethyl)-4,8,13,18-tetramethyl-20-oxo-3-phorbinepropanoic acid (**1**) has not been described as a natural compound before but has been previously obtained through organic synthesis [20,21,22]. Synthetic porphyrins, along with related chlorin compounds, have a long history of application as photosensitizers in PDT. PDT has been initially developed and approved to treat bladder, esophageal, and lung forms of solid cancers, and basal cell carcinoma [6]. PDT since then has found additional applications, e.g., in dermatology [23].

PDT worldwide is a medical market worth some USD 3.4 billion in 2019, projected to grow to USD 11.9 billion by 2027 (verifiedmarketresearch.com/product/photodynamic-therapy-market/) and relies on the use of a limited number of approved photosensitizers, such as Photofrin^®^ and Verteporfin^®^ in the US, Foscan^®^, and Tookad^®^ in Europe, Photolon^®^ and Photosens^®^ in Russia, Talaporfin sodium^®^ in Japan, and Hemoporfin^®^ in China [24]. Complicated synthetic routes contributing to the high cost of the photosensitizers are among the factors for the constant search for novel PDT compounds [25]. In our work, we describe efficient isolation from an abundant brittle star species of a cytotoxic porphyrin with strong absorption in the red spectrum. If medical usefulness is proven, upscaling of the isolation of this potential natural source photosensitizer can be developed, accompanied by a mariculture of *O. sarsii* with approaches developed for other North Pacific invertebrates [26]; biotechnological production or optimization of the chemical synthesis is also conceivable. Thus, our discovery may pave the way for novel natural-based developments of targeted anticancer therapies.

## 3. Materials and Methods

### 3.1. Raw Material, Homogenization, and Extraction

Animal samples of *Ophiura sarsii* were collected at the depths of 15–18 m in the Bogdanovich Bay near Cape Vyatlin on Russky Island in the Peter the Great Gulf, Sea of Japan, in May 2019. The exact coordinates of the catch were 42°57′48.9″ N 131°54′24.8″ E. Sample collection was performed by the standards approved by the Ministry of Education and Science (Russia); all efforts were made to minimize animal suffering. Freshly caught animals were washed twice under running water, after which they were immediately frozen and stored at −80 °C.

Of wet *O. sarsii*, 120 g were homogenized using a vibrating mill (Retsch, Éragny, France) in heptane at a vibration frequency of 1500 rpm for 2 min until a homogeneous gelatinous mass was obtained. Further, the crushed sample was extracted with 500 mL of solvents. Extraction was performed sequentially with four solvents of increasing polarity: heptane, chloroform, ethanol, and water, at room temperature for 8–12 h. Each extract was then filtered using Whatman 47 mm GF/C glass fiber filter (Sigma-Aldrich, St. Louis, CO, USA). Each filtrate was concentrated in a rotary evaporator. The yield of the obtained extracts was as follows: heptane 38.2 mg, chloroform 64.8 mg, ethanol 509 mg, and water 513.5 mg.

### 3.2. Dual Luciferase Assay

The assay was performed as described previously [14,27]. The TNBC cell line BT-20 stably transfected with the TopFlash reporter was seeded in DMEM medium supplemented with 10% FBS (both Gibco, Gaithersburg MD, USA) into a 384-well white flat bottom plate for luminescence analysis (Greiner, Monroe NC, USA) at a density of 6000 cells/well in a final volume of 20 μL of the medium.

The next day, the medium was removed using plate washer (Biotek, Winooski VT, USA) and the attached cells in the plate were transfected with the plasmid encoding the Renilla reniformis luciferase gene under control of the CMV promoter. The procedure was done in accordance with the manufacturer’s instructions using in total 300 μL of Opti-MEM medium per plate (Gibco, USA) with 12 μg/mL of DNA and 40 μL/mL of X-tremeGENE reagent (Roche, Basel, Switzerland) in the final volume of Opti-MEM/DMEM mixture of 20 µL/well.

Upon overnight incubation, the transfection media was removed using the plate washer and the samples from *O. sarcii*, provided as DMSO stocks, were added to the reporter cells in a series of dilutions in DMEM starting from 100 μg/mL for extracts and from 25 μg/mL for the pure substance (based on dry weight). To activate the Wnt cascade in the BT-20-TopFlash reporter cells, Wnt3a protein purified as described earlier [28] was added to the final concentration of 2.5 µg/mL. The final concentration of DMSO in the wells was maintained at 0.5%. Each experiment was carried out in four replicates. Cells with substances were incubated overnight at 37 °C, after which the luminescence was recorded by the method of the dual luciferase analysis, below.

To measure the activity of luciferases, the culture medium was completely removed from all wells of the plate. Next, the luciferase activity of the firefly and *Renilla* luciferases was detected sequentially in individual wells of a 384-well plate through injection of corresponding measurement solutions [29] in the Infinite M Plex multifunctional plate reader with the injection module (Tecan, Männedorf, Switzerland).

The activity of luciferases was recalculated in % relative to the cells in the medium supplemented with Wnt3a alone, which served as the positive control. Statistical processing of the results, and plotting of graphs, was performed using Prism 8 (GraphPad Software, San Diego, CA, USA).

### 3.3. General Experimental Procedures

UV spectra were recorded on an Agilent Cary 60 UV-Vis (Santa-Clara, CA, USA) in MeOH. NMR data were recorded on a Bruker Avance Neo 600 MHz NMR spectrometer equipped with a QCI 5 mm cryoprobe and a SampleJet automated sample changer (Bruker BioSpin, Rheinstetten, Germany). 1D and 2D NMR experiments (^1^H, COSY, ROESY, and HSQC) were recorded in DMSO-*d*_6_. Chemical shifts are reported in parts per million (δ) and coupling constants (*J*) in Hz. The residual DMSO-*d*_6_ signal (δ_H_ 2.50; δ_C_ 39.5) were used as internal standards for ^1^H and ^13^C NMR, respectively. HRMS data were obtained on a Waters Acquity UHPLC system interfaced to a Q-Exactive Focus mass spectrometer (Thermo Scientific, Bremen, Germany), using a heated electrospray ionization ((HESI-II) source). The biotransformation reactions were monitored on a UHPLC-PDA-ELSD-MS (Waters) instrument equipped with a single quadrupole detector using heated electrospray ionization. Analytical HPLC was carried out on an HP 1260 Agilent system equipped with a photodiode array detector (Agilent Technologies, Santa Clara, CA, USA). Semipreparative HPLC was conducted on a Shimadzu system equipped with an LC-20 A module pumps, an SPD- 20 A UV/VIS, a 7725I Rheodyne^®^ valve, and an FRC-40 fraction collector (Shimadzu, Kyoto, Japan). Before analysis, chloroform, ethanol, and water extracts were filtered through SPE C_18_ cartridges (Finisterre 1000 mg/6 mL, Teknokroma, Barcelona, Spain). Cartridges were first activated by eluting 10 mL of MeOH, then conditioned with 10 mL of H_2_O. The extract was solubilized in water with a minimum amount of MeOH and loaded on the cartridge. Around 10 mL of water were first eluted to collect a first fraction, followed by 10 mL of MeOH.

### 3.4. UHPLC-PDA-ELSD-MS Analysis

The extracts, fractions, and pure compounds were analyzed on an Ultra-high-performance liquid chromatography system equipped with a photodiode array, an evaporative light-scattering detector, and a single quadrupole detector using heated electrospray ionization (UHPLC-PDA-ELSD-MS) (Waters, Milford, MA, USA). The ESI parameters were the following: capillary voltage 800 V, cone voltage 15 V, source temperature 120 °C, and probe temperature 600 °C. Acquisition was done both in the positive and negative ionization mode (PI and NI) with an m/z range of 150−1000 Da. The chromatographic separation was performed on an Acquity UPLC BEH C_18_ column (50 × 2.1 mm i.d., 1.7 μm; Waters) at 0.6 mL/min, 40 °C with H_2_O (A) and MeCN (B) both containing 0.1% formic acid as solvents. The gradient was carried out as follow: 5–100% B in 7 min, 1 min at 100% B, and a re-equilibration step at 5% B in 2 min. The ELSD temperature was fixed at 45 °C, with a gain of 9. The PDA data were acquired in from 190 to 500 nm, with a resolution of 1.2 nm. The sampling rate was set at 20 points/s.

### 3.5. UHPLC-HRMS Analysis

The extracts, fractions, and pure compounds were profiled on a Waters Acquity UHPLC system equipped with a Q-Exactive Focus mass spectrometer (Thermo Scientific, Bremen, Germany), using heated electrospray ionization source (HESI-II). The chromatographic separation was carried out on an Acquity UPLC BEH C_18_ column (50 × 2.1 mm i.d., 1.7 μm; Waters) at 0.6 mL/min, 40 °C with H_2_O (A) and MeCN (B) both containing 0.1% formic acid as solvents. The gradient was carried out as follow: 5–100% B in 7 min, 1 min at 100% B, and a re-equilibration step at 5% B in 2 min. The ionization parameters were the same used in [30].

### 3.6. Semipreparative HPLC-UV Purification of the Butanol Fraction

Before the fractionation process, liquid–liquid extraction was performed on the ethanol extract to remove salts. The extract (509 mg) was solubilized in 30 mL of butanol and 50 mL of water and transferred to a separatory funnel. The organic phase (butanol phase) was extracted 3 times with 50 mL of water. Each aqueous phase was analyzed by UHPLC-PDA-ELSD-MS. The third aqueous phase contained almost no detectable peaks, indicating an efficient salt removal. The butanol phase (212 mg) was dried on a rotary evaporator. For fractionation of this butanol phase by semipreparative HPLC, the chromatographic separation was optimized on an analytical HPLC HP 1260 Agilent system equipped with a photodiode array detector (Agilent Technologies, Santa Clara, CA, USA). The separation was done on a Waters X-Bridge C_18_ column (250 mm × 4.6 mm i.d., 5 µm) at 1 mL/min, 25 °C with H_2_O (A) and MeOH (B) both containing 0.1% formic acid as solvents. The gradient was carried out as follows: 80–100% B in 60 min, followed by 10 min at 100% B. These optimized HPLC analytical conditions were geometrically transferred by gradient transfer to the semi-preparative HPLC scale [31]. Of the extract 50 mg was fractionated using reverse-phase semipreparative HPLC-UV on a Shimadzu system equipped with an LC-20 A module pumps, an SPD-20 A UV/VIS, a 7725I Rheodyne^®^ valve, and an FRC-40 fraction collector (Shimadzu, Kyoto, Japan). The separation was performed on a Waters X-Bridge C_18_ column (250 mm × 19 mm i.d., 5 μm) equipped with a Waters C_18_ precolumn cartridge holder (10 mm × 19 mm i.d., 5 μm) at 17 mL/min, 25 °C with H_2_O (A) and MeOH (B) both containing 0.1% formic acid as solvents. The gradient was carried out as follow: 80–100% B in 60 min, followed by 10 min at 100% B, with UV detection at 210 and 366 nm. The extract was injected on the semipreparative HPLC column using a dry load methodology recently developed in our laboratory [32]. The separation yielded 51 fractions. Each fraction was evaluated for cytotoxic activity and submitted to UHPLC-PDA-ELSD-MS analysis chemical content assessment. Among the fractions obtained, fraction 14 afforded compound **1** (3 mg) with interesting biological activities.

### 3.7. Structure Elucidation of Compound **1**

(3*S*,4*S*)-14-ethyl-9-(hydroxymethyl)-4,8,13,18-tetramethyl-20-oxo-3-phorbinepropanoic acid (**1**). [α]^20^_D_ +350 (*c* 0.004, MeOH); UV (MeOH) λ_max_ (log ε) 204 (3.90), 293 (sh) (3.54), 409 (3.45), 663 (2.63) nm; ^1^H-NMR (DMSO, 600 MHz) δ 1.64 (3H, t, *J* = 7.6 Hz, H_3_-8b), 1.77 (3H, d, *J* = 7.4 Hz, H_3_-18a), 2.11 (1H, m, H-17a”), 2.32 (1H, m, H-17b”), 2.55 (1H, m, H-17b’), 2.64 (1H, m, H-17a’), 3.25 (3H, s, H_3_-7a), 3.39 (3H, s, H_3_-2a), 3.63 (3H, s, H_3_-12a), 3.73 (2H, q, *J* = 7.6 Hz, H_2_-8a), 4.32 (1H, dt, *J* = 10.0, 2.9 Hz, H-17), 4.58 (1H, qd, *J* = 7.4, 2.9 Hz, H-18), 5.12 (1H, d, *J* = 19.4 Hz, H-13b”), 5.23 (1H, d, *J* = 19.4 Hz, H-13b’), 5.74 (2H, s, H_2_-3a), 8.83 (1H, s, H-20), 9.57 (1H, s, H-5), 9.75 (1H, s, H-10); ^13^C-NMR (DMSO, 150 MHz) δ 10.3 (C-2a), 10.5 (C-7a), 11.2 (C-12a), 17.1 (C-8b), 18.3 (C-8a), 22.5 (C-18a), 29.0 (C-17a), 30.4 (C-17b), 47.1 (C-13b), 48.9 (C-18), 50.6 (C-17), 53.9 (C-3a), 93.2 (C-20), 96.7 (C-5), 103.9 (C-10); HR-ESI/MS analysis: *m/z* 539.2644 [M + H]^+^, (calcd for C_32_H_35_N_4_O_4_, 539.2653, Δ = 1.6 ppm).

### 3.8. MTT Survival Assay in Breast Cancer Cells

Breast cancer cells BT-20, HCC 1395, HCC 1806, MDA-MB-231, MDA-MB-468, and MCF-7 were cultured in DMEM medium supplemented with 10% FBS (Gibco, USA). Non-tumorigenic epithelial breast cells MCF-10A were cultured in DMEM/F12 medium (Gibco, Waltham, MA, USA) supplemented with 10% FBS. For analysis, the cells were seeded in transparent 384-well plates (VWR, Dietikon, Switzerland) at the following densities: HCC 1395, HCC 1806—2000 cells/well; BT-20, MDA-MB-231, MDA-MB-468—3000 cells/well; and MCF-10A—5000 cells/well.

A porphyrin sample was redissolved with DMSO and added to the cells at indicated concentrations starting from 25 μg/mL. The final concentration of DMSO in the wells was maintained at 0.5%. Each experiment was done in four repeats. Cells were incubated for three days at 37 °C, after which their numbers were quantified, as follows.

The medium was removed from all wells of the plate and 20 μL of the MTT (triazolyltetrazoliumumbromide) reagent (Sigma-Aldrich, USA) in PBS at a concentration of 0.5 mg/mL was added and incubated for 3.5 h at 37 °C. The solution was removed from all wells of the plate using the plate washer. Subsequently, 50 μL of DMSO was added, incubated for 5 min, and the optical density of the solution was measured at the wavelength of 590 nm by the Infinite M Plex multifunctional plate reader (Tecan, Switzerland). Statistical processing of the results, and plotting of graphs, was performed using Prism 8.

## Figures and Tables

**Figure 1 marinedrugs-19-00011-f001:**
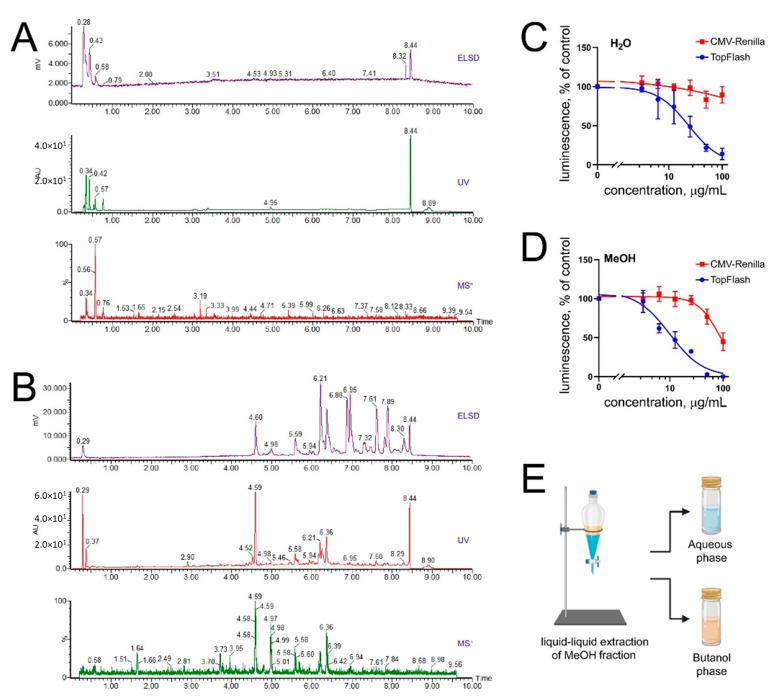
Extracts of *Ophiura sarsii* reveal anti-Wnt and cytotoxic activities. (**A**,**B**) Chromatograms of ethanol extracts of *O. sarsii* after additional solid-phase extraction (SPE) with elution with water (**A**) and methanol (**B**). Polar compounds, most probably inorganic salts, are abundant in the water fraction (**A**), while the methanol fraction (**B**) appears enriched in secondary metabolites of interest. (**C**,**D**) Acute cytotoxicity (CMV-Renilla) and anti-Wnt (TopFlash) biological assays reveal that the methanol fraction (**D**) is indeed enriched in both activities. (**E**) Scheme of the liquid–liquid extraction of the methanol fraction, which produced the butanol phase where the biological activities concentrated.

**Figure 2 marinedrugs-19-00011-f002:**
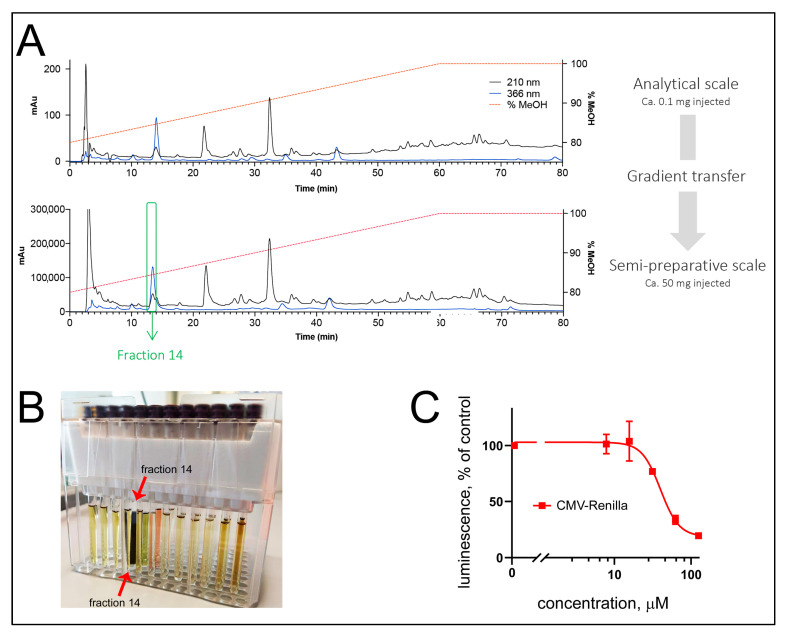
Isolation of the cytotoxic activity from *O. sarsii*. (**A**) Gradient transfer from the analytical HPLC-PDA (up) to the semipreparative HPLC-UV (down) of the butanol phase, resulted in the isolation of fraction 14. (**B**) NMR tubes of the different fractions, showing the strong dark-blue coloration of fraction 14. (**C**) Fraction 14 mediated the cytotoxic activity of *O. sarsii* in the acute cytotoxicity (CMV-Renilla) assay.

**Figure 3 marinedrugs-19-00011-f003:**
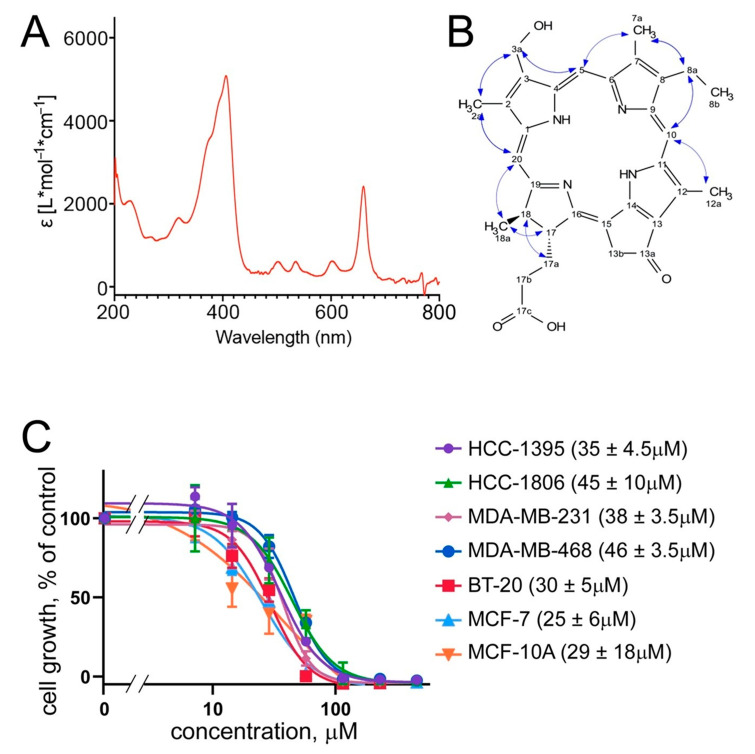
A porphyrin compound, (3*S*,4*S*)-14-Ethyl-9-(hydroxymethyl)-4,8,13,18-tetramethyl-20-oxo-3-phorbinepropanoic acid, mediates cytotoxicity of *O. sarsii*. (**A**) UV spectrum of the fraction 14 reveals absorbance peaks typical for porphyrins. (**B**) NMR identified the compound in fraction 14 as (3*S*,4*S*)-14-Ethyl-9-(hydroxymethyl)-4,8,13,18-tetramethyl-20-oxo-3-phorbinepropanoic acid (**1**); ROESY correlations of compound **1** are shown as blue arrows. (**C**) Cell proliferation (MTT) assay shows a broad cytotoxicity of compound **1** against breast cancer lines. IC_50_ data are shown to the right of the graph as mean ± SEM values, *n* = 4.

## Data Availability

The data presented in this study are fully available in the main text and supplementary materials of this article.

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
