# Peer review of "A Cytotoxic Porphyrin from North Pacific Brittle Star Ophiura sarsii"

_marinedrugs, 2020, doi:10.3390/md19010011_

Round 1

Reviewer 1 Report

This manuscript describes the preparation of extracts obtained from the North Pacific brittle star Ophiura sarcii in the series of increase polarity extractions. Ethanol exact was then separated by semi-preparative HPLC and one of the fractions appeared to be rich in specific porphyrin derivative, for which the cytotoxicity was measured for number of cell lines including triple-negative breast cancer. Although this compound has been chemically synthetized before this is an interesting new report, how it can be obtained with relatively easy procedure from the natural source. Unfortunately presented data suggest similar toxicity towards cancerous and non-cancerous (MCF-10A) cell line. Also the comparison of extracted with synthetic porphyrins would be a valuable addition to this paper, but in the reviewer’s opinion this manuscript can be published as is. Data (NMR, MS) supports the claim, manuscript is well written, clear to read and understand.

Author Response

We sincerely thank the Reviewer for his/her constructive assessment and positive evaluation of our manuscript. We highly appreciate the conclusion of the Reviewer that the manuscript can be published as is. Nevertheless, we polish once again the English of our manuscript (as the Reviewer noted that some minor spell check would be good). Further, we stress once again in the Discussion that for the photodynamic therapy (PDT), the photosensitizer does not have to demonstrate a better toxicity towards cancer cells, as the targeted nature of the PDT is achieved through focal excitation of the tissue.

Reviewer 2 Report

This paper describes the results of an impressive collaboration between scientists at the Far Eastern Federal University in Vladivostok and the University of Geneva, on the identification and cytotoxic activity of a novel photosensitive compound in the North Pacific Brittle Star.

I would like to request the authors to respond to the following questions.

What is the potential physiological role of this compound in the organism?

Since the compound also shows cytotoxicity in the non-cancerous MCF-10A cells, how do the authors envision its use as a cancer therapeutic?

Other points:

The titles of the references should be lower case, and the journal titles should be uniformly abbreviated.

The units should be separated from the numerical values by a space.

"such as the colon cancer" -> "such as colon cancer"

"the triple-negative breast cancer" -> "triple-negative breast cancer" 

 Please clarify "dedicated assays"

"long-wave tumor illumination" -> "long-wavelength tumor illumination"

 FCS -> FBS

"Next day" - "The next day"

"in accordance with manufacturer’s" -> "in accordance with the manufacturer’s" 

What is the source of Prism 8?

In ref 10, Ophiura sarsii should be italicized

Author Response

We sincerely thank the Reviewer for his/her constructive assessment of our manuscript and the overall highly positive evaluation of it. The revised manuscript addresses all the minor issues raised by the Reviewer, as detailed below.

I would like to request the authors to respond to the following questions.

What is the potential physiological role of this compound in the organism?

We have added a sentence in the discussion to address this issue, as follows:

Subsequent studies will address whether (3S,4S)-14-ethyl-9-(hydroxymethyl)-4,8,13,18-tetramethyl-20-oxo-3-phorbinepropanoic acid is synthesized by O. sarsii itself, or whether the porphyrin in these brittle stars is an outcome of their dietary preferences. In the latter case, it will be worth investigating whether such dietary acquisition (with potential subsequent in-host modification) of the porphyrin is a local phenomenon, or whether it is as wide-spread as the northern hemisphere areal of O. sarsii. It is worth-mentioning that a similar porphyrin, pyropheophorbide a methyl ester, has been described in the red tide dinoflagellates Heterocapsa circularisquama [18]. In the former case, subsequent studies will be directed to address the question of the physiological role the porphyrin plays in O. sarsii, which might be linked e.g. with a protective function of the compound.

Since the compound also shows cytotoxicity in the non-cancerous MCF-10A cells, how do the authors envision its use as a cancer therapeutic?

To stress this important issue, which was already addressed in the Introduction, we have added the following sentences to the Discussion:

Noteworthy, the cytotoxicity of compound 1 against cancerous and non-cancerous breast epithelial cells is similar (Figure 3C). This is typical for the photosensitizers applied in the photodynamic therapy (PDT). In PDT, the targeted nature of the treatment is achieved not through selective toxicity of compounds against cancerous cells. Instead, in PDT, tissues are preloaded with the photosensitizer (ideally absorbing in the red-shifted part of the spectrum that penetrates deeper in biological tissues) either systemically or topically, which is followed by targeted laser-based illumination with the correct wavelength. As a result, photosensitizer is activated, and emits oxygen radicals, toxic for the cells [5, 6].

The titles of the references should be lower case, and the journal titles should be uniformly abbreviated.

Done.

The units should be separated from the numerical values by a space.

Done.

"such as the colon cancer" -> "such as colon cancer"

Done.

"the triple-negative breast cancer" -> "triple-negative breast cancer" 

Done.

Please clarify "dedicated assays"

Substituted to "set of assays monitoring Wnt signaling in the cancer cells"

"long-wave tumor illumination" -> "long-wavelength tumor illumination"

Done.

FCS -> FBS

Done.

"Next day" - "The next day"

Done.

"in accordance with manufacturer’s" -> "in accordance with the manufacturer’s" 

Done.

What is the source of Prism 8?

Information added (GraphPad Software, USA).

In ref 10, Ophiura sarsii should be italicized

Done.

Reviewer 3 Report

The manuscript by Antonina Klimenko et al. describes the identification of a new porphyrin with cytotoxic properties. The substance was isolated from the North Pacific Brittle Star.

In the introduction, the authors give a good overview of the importance of cytotoxic substances that can be obtained from marine organisms, especially invertebrates. Porphyrins can be used as photosensitizers in the photodynamic tumour therapy of solid tumours. Thus, great importance can be attached to the research of this complex group of active substances.

In the manuscript, the molecular analysis of the new porphyrin was presented in detail. In addition to the extraction from the marine organism, the structure of the target substance was described by means of HPLC and different elution methods using HRMS and 2D NMR. All steps were described in detail and comprehensibly.

In addition to the analysis, the cytotoxic effect of the new substance on various breast cancer cell lines was investigated by means of MTT assays.

  1. The substance was isolated from the whole organism. Would it be possible to assign the origin to an organ system of Ophiura sarsii?
  2. The authors mentioned that the existence of porphyrin could also be caused by the ingestion of, among others, unicellular dinoflagellates. Is it possible to clearly delineate the origin of the substance ?
  3. The complexity of primary analysis and identification should be summarised in a manuscript. The cell biological and cytotoxic work should be summarised and presented in detail in a second manuscript.
  4. the economic importance of PDT (line 161) should not be mentioned.
  5. Figure 3c is confusing and should be revised.

Author Response

We sincerely thank the Reviewer for his/her constructive assessment of our manuscript and for the overall positive evaluation of it. 

The revised version of the paper addresses the issues raised by this and the other two Reviewers. Below I provide the responses to the individual points raised by this Reviewer.

  1. The substance was isolated from the whole organism. Would it be possible to assign the origin to an organ system of Ophiura sarsii?
  2. The authors mentioned that the existence of porphyrin could also be caused by the ingestion of, among others, unicellular dinoflagellates. Is it possible to clearly delineate the origin of the substance ?

We have now added some more discussion on these points, which reads as follows:

Porphyrins have been identified in several marine living organisms including Echinodermata [13]. However, they were not seen – and were considered absent – in the class of Ophiuroidea [11, 12]. Subsequent studies will address whether (3S,4S)-14-ethyl-9-(hydroxymethyl)-4,8,13,18-tetramethyl-20-oxo-3-phorbinepropanoic acid is synthesized by O. sarsii itself, or whether the porphyrin in these brittle stars is an outcome of their dietary preferences. In the latter case, it will be worth investigating whether such dietary acquisition (with potential subsequent in-host modification) of the porphyrin is a local phenomenon, or whether it is as wide-spread as the northern hemisphere areal of O. sarsii. It is worth-mentioning that a similar porphyrin, pyropheophorbide a methyl ester, has been described in the red tide dinoflagellates Heterocapsa circularisquama [18]. In the former case, subsequent studies will be directed to address the question of the physiological role the porphyrin plays in O. sarsii, which might be linked e.g. with a protective function of the compound.

In brief, we believe it is the subject of subsequent works to provide a detailed answer to this question.

3. The complexity of primary analysis and identification should be summarised in a manuscript. The cell biological and cytotoxic work should be summarised and presented in detail in a second manuscript.

We thank the Reviewer for this suggestion that our work might be worth of two separate publications. However, also given the fact that the other two Reviewers see the great value of our story in the unity of the primary analysis and cellular investigation, we would politely keep the structure of our manuscript as is.

4. The economic importance of PDT (line 161) should not be mentioned.

I understand that the Reviewer wishes to see the scientific value of our discovery to be separated from it economic potential. However, we (and the two other Reviewers) think that the brief mentioning of the economic importance of the photodynamic therapy is a worthy bit of information to be added to the Discussion of our paper.

5. Figure 3c is confusing and should be revised.

In order to improve the clarity of figure 3c, we have expanded the legend of it, which now reads as follows:

(C) Cell proliferation (MTT) assay shows a broad cytotoxicity of compound 1 against breast cancer lines. IC50 data are shown to the right of the graph as mean ± sem values, n=4.